# ChebLieNet: Invariant Spectral Graph NNs Turned Equivariant by Riemannian Geometry on Lie Groups

## Abstract

We introduce ChebLieNet, a group-equivariant method on (anisotropic) manifolds. Surfing on the success of graph- and group-based neural networks, we take advantage of the recent developments in the geometric deep learning field to derive a new approach to exploit any anisotropies in data. Via discrete approximations of Lie groups, we develop a graph neural network made of anisotropic convolutional layers (Chebyshev convolutions), spatial pooling and unpooling layers, and global pooling layers. Group equivariance is achieved via equivariant and invariant operators on graphs with anisotropic left-invariant Riemannian distance-based affinities encoded on the edges. Thanks to its simple form, the Riemannian metric can model any anisotropies, both in the spatial and orientation domains. This control on anisotropies of the Riemannian metrics allows to balance equivariance (anisotropic metric) against invariance (isotropic metric) of the graph convolution layers. Hence we open the doors to a better understanding of anisotropic properties. Furthermore, we empirically prove the existence of (data-dependent) sweet spots for anisotropic parameters on CIFAR10. This crucial result is evidence of the benefice we could get by exploiting anisotropic properties in data. We also evaluate the scalability of this approach on STL10 (image data) and ClimateNet (spherical data), showing its remarkable adaptability to diverse tasks.

## 1 Introduction

Deep learning is a class of machine learning algorithms inspired by the human brain's network of neurons [Goodfellow et al., 2016]. These algorithms use a hierarchical structure of neural layers to extract higher-level features from the raw input progressively. In the past few years, the growing computational power of modern GPU-based computers and the availability of large training datasets in the field of machine learning have made it possible to successfully train neural networks with many layers and degrees of freedom. Consequently, deep learning has revolutionized many machine learning tasks in recent years, ranging from image and video processing to speech recognition and natural language understanding.

Many neuroscientific research results served as focal points in the development of deep learning algorithms. When Hubel and Wiesel [1962] studied the visual cortex in the brain, they made three important discoveries. First, they observed a one-to-one correspondence between spatial locations in the retina and neurons in the brain that fired as a response to line-like visual stimuli. Second, the activity of the neurons changed depending on the orientation of the line, uncovering a neat organization based on local orientations. Last, the neurons sometimes fired only when the line was moving in a particular direction. Later, Bosking et al. [1997] showed that neurons that are aligned fire together, indicating the presence of a type of long-range interactions. All these results motivated the development of a mathematical framework for modeling visual perception based on sub-Riemannian

geometry on the space of positions and orientations, which is typically modeled with the Lie group SE(2) [Petitot, 2003, Citti and Sarti, 2006, Duits et al., 2014]. Apart from the neurophysiological inspiration, group equivariance has also been proven to be an excellent inductive bias [Cohen and Welling, 2016] not only in computer vision (as the translation equivariance property of CNNs as shown) but also in physics [Finzi et al., 2020] and molecular data analysis [Fuchs et al., 2021, Jumper et al., 2020]. In this work, we propose to build group equivariant graph neural networks via the same principle that underlie the sub-Riemannian, neurogeometrical modeling of the visual cortex.

Our work connects the observations by Hubel and Wiesel [1962] and Bosking et al. [1997] on two levels. First, the organization of visual data based on their location and orientation [Hubel and Wiesel, 1962] is modeled by Lie group convolutions [Bekkers, 2019], in which feature maps encode response for every position and every orientation. Second, long-range interactions between aligned neurons [Bosking et al., 1997] are modeled by building graphs with affinity matrices based on (approximate) sub-Riemannian distances on the Lie groups, inspired by sub-Riemannian image analysis methods such as [Franken and Duits, 2009, Bekkers et al., 2015, Favali et al., 2016, Mashtakov et al., 2017, Boscain et al., 2018, Duits et al., 2018, Baspinar et al., 2021].

Defferrard et al. [2020] showed how to construct powerful graph NNs that are faithful to the manifolds on which they are defined. Nevertheless, the layers themselves are based on rotationally invariant (Laplacian) convolutions. In order to exploit directional cues in the data, group convolutions are desirable [Cohen et al., 2018, Kondor and Trivedi, 2018, Cohen and Welling, 2016, Bekkers, 2019]. However, since Laplacian operators are intrinsically isotropic, there is no point applying them to the lifted feature maps on the group unless we construct anisotropic metrics on the groups. Therefore, we adopt the Lie group viewpoint by Sanguinetti et al. [2015] to define anisotropic Riemannian metrics based on left-invariant vector fields on the group. Once an anisotropic Riemannian graph is constructed, any spectral method can directly be applied to this graph. The resulting graph neural networks will then, by construction, be equivariant and capable of utilizing directional cues in data.

Before going further into the details, we summarize our main contributions:

- We introduce ChebLieNet, an equivariant graph Laplacian-based neural network based on Lie groups equipped with an anisotropic Riemannian metric.

- The Riemannian geometry is automatically derived from a standard base space (e.g. $\mathbb{R}^2$ or the sphere), which makes our approach flexible and effective in building group equivariant graph neural networks for a variety of data structures (e.g. 2D and spherical data).

- We demonstrate the equivariance property of ChebLieNet, both in theory and in practice. This property guarantees that the neural network's predictions are robust against given transformations, which is not necessarily the case with methods based on data augmentation.

- We show that the use of directional information via anisotropic Riemannian spaces could benefit many tasks.

- We show the flexibility of the method by considering two different problems; we validate on classification problems with 2D image data and a segmentation problem on spherical data via the construction of a sub-Riemannian geometry on $SE(2)$ and $SO(3)$ respectively.

## 2 Related works

### 2.1 Group equivariant convolutional neural networks

Deep convolutional neural networks [LeCun et al., 1995] have proven to be compelling models for pattern recognition tasks on images, video, and audio data. Although a robust theory of neural network design is currently lacking, a large amount of empirical evidence supports the notion that both convolutional weight sharing, depth, and width are essential for good predictive performance. Such properties are enabled through the equivariance property of convolutions (convolving a shifted image is the same as translating its result).

Lenc and Vedaldi [2015] showed that the AlexNet CNN Krizhevsky et al. [2012] trained on ImageNet learns representations equivariant to flips, scalings, and rotations spontaneously. This supports the idea that equivariance is an excellent inductive bias for deep convolutional networks. In the last few years, a joint effort has been made to build group equivariant networks. By the introduction of group

convolutions in deep learning, Cohen and Welling [2016] generalize the translation equivariance property to larger groups of symmetries, including rotations and reflections. Kondor and Trivedi [2018] gave a rigorous, theoretical treatment of convolution and equivariance in neural networks concerning any compact group's action. One of the main contributions of that work was to show that, given some natural constraints, the convolutional structure is not just a sufficient but also a necessary condition for equivariance to a compact group's action. In a similar spirit, in [Bekkers, 2019] it is shown that any bounded linear operator is equivariant to Lie groups if and only if it is a group convolution. In our work, we propose to build group equivariant neural networks via left-invariant Laplace operators on Lie groups, which indeed can be seen as group convolutions with kernels that are the fundamental solutions of the Laplace operator. The result is a Lie group equivariant Chebyshev-type neural network [Defferrard et al., 2016] that we will refer to as ChebLieNet.

## 2.2 Graph neural networks

Using the term geometric deep learning, Bronstein et al. [2017, 2021] give an overview of deep learning methods in the non-Euclidean domain, including graphs and manifolds. They present different examples of geometric deep learning problems and available solutions, fundamental difficulties, applications, and future research directions in this nascent field.

One of the main challenges when working with graph data it to deal with the inter-dependencies between points. Indeed, the derivations of most standard machine learning models firmly base on an independence assumption. For this reason, transferring existing methods on a graph appears doomed to failure, and it seems necessary to build models acting directly on graphs. Due to its success on Euclidean data, the development of a convolution-like operator on graphs has been largely studied. Because the notion of space is not naturally defined on a graph, we lack a straightforward generalization of the convolutional operator from grid data to graphs [Scarselli et al., 2008, Bruna et al., 2013, Henaff et al., 2015, Defferrard et al., 2016, Kipf and Welling, 2016, Masci et al., 2015, Boscaini et al., 2016, Monti et al., 2017].

Spectral approaches have a solid mathematical foundation in graph signal processing. Rather than using the traditional spatial definition of the convolution, it proposes to see this operation from a spectral perspective. Based on the convolution theorem, it defines the convolution operator from the graph spectral domain via the eigendecomposition of the graph Laplacian (see App. A.3).

**Definition 2.1 (Spectral graph convolution)** *Let $\mathcal{G} = (\mathcal{V}, \mathcal{E}, \boldsymbol{W})$ be a graph with Laplacian $\hat{\boldsymbol{\Delta}}$ and let $f$ and $g$ be two functions defined on $\mathcal{V}$. We define the $\mathcal{G}$-convolution $*_{\mathcal{G}}$ of $f$ and $g$ as:*

$$f *_{\mathcal{G}} g = \boldsymbol{\Phi}(\hat{\boldsymbol{g}} \odot \hat{\boldsymbol{f}}) = \boldsymbol{\Phi}(\boldsymbol{\Phi}^{\top} \boldsymbol{g} \odot \boldsymbol{\Phi}^{\top} \boldsymbol{f}), \tag{1}$$

*with eigenvectors $\boldsymbol{\Phi}$ obtained through the unique eigendecomposition $\hat{\boldsymbol{\Delta}} = \boldsymbol{\Phi}\boldsymbol{\Lambda}\boldsymbol{\Phi}^{T}$.*

While this definition alleviates the difficulty of deriving a convolution operator in the spatial domain, other difficulties arise. First of all, because the Laplacian of a graph is an intrinsic operator, it is domain-dependent, and the spectral-convolution is too. It implies that a model built on this framework cannot be easily transferred from a graph to another as expressed in a different "language". Nevertheless, this is not a problem for us since we are focusing on fixed manifold graphs. Next, there is no guarantee that filters represented in the spectral domain are spatially localized. Henaff et al. [2015] successfully bypassed this problem by defining smooth spectral filter coefficients, arguing that if spectral filters are smooth, they are spatially localized. Last but not least, the Laplacian's eigendecomposition makes the method expensive in terms of memory and time. Indeed, the forward and inverse graph Fourier transforms (via $\boldsymbol{\Phi}^{T}$ and $\boldsymbol{\Phi}$) incur expensive multiplications as no FFT-like algorithm exists on general graphs. Defferrard et al. [2016] alleviated the cost of explicitly computing the graph Laplacian using spatially-localized filters with Chebyshev polynomials.

**Definition 2.2 (Chebyshev convolutional layer)** *Let $\mathcal{G} = (\mathcal{V}, \mathcal{E}, \boldsymbol{W})$ be a graph with rescaled Laplacian[1] $\tilde{\boldsymbol{\Delta}}$, $\boldsymbol{x} \in \mathbb{R}^{|\mathcal{V}| \times d_i}$ be an input features' vector and $\Theta_j \in \mathbb{R}^{d_i \times d_o}$ learnable filters. The*

---

[1]Because Chebyshev polynomials are defined in the range $[-1, 1]$, it is necessary to rescale the graph Laplacian with $\tilde{\boldsymbol{\Delta}} = 2\lambda_{\max}^{-1}\hat{\boldsymbol{\Delta}} - \boldsymbol{I}$ where $\lambda_{\max}$ is the largest eigenvalue of $\hat{\boldsymbol{\Delta}}$.

*output features' vector $\boldsymbol{y} \in \mathbb{R}^{|\mathcal{V}| \times d_o}$ is computed as:*

$$\boldsymbol{y} = \sum_{j=0}^{R-1} \boldsymbol{z}_j \boldsymbol{\Theta}_j \quad with \quad \boldsymbol{z}_0 = \boldsymbol{x}, \quad \boldsymbol{z}_1 = \tilde{\boldsymbol{\Delta}} \boldsymbol{x} \quad and \quad \boldsymbol{z}_j = 2\tilde{\boldsymbol{\Delta}} \boldsymbol{z}_{j-1} - \boldsymbol{z}_{j-2}. \quad \forall j \geq 2. \quad (2)$$

Kipf and Welling [2016] simplified this formulation a bit by considering the construction of single-parametric filters that are linear with relation to $\tilde{\boldsymbol{\Delta}}$. They further approximate $\lambda_{\max} \simeq 2$ as they expect that neural network parameters will adapt to this change in scale during training.

## 3   Method

Our method can be seen as an extension of the original ChebNet [Defferrard et al., 2016, Perraudin et al., 2019]. Instead of directly working on a homogeneous base space, we first extend it to a higher dimensional space (Lie group). The goal of this extension is to convert the previously invariant spectral convolutional layers into equivariant layers.[2]

### 3.1   Anisotropic manifold graph

In order to define the anisotropic manifold graphs we have to consider two types of manifolds. The base manifold $\mathcal{M}$ and a Lie group $G$ that acts transitively on $\mathcal{M}$. The latter implies that $\mathcal{M}$ is a homogeneous space of $G$, which means that any two points $m_1, m_2 \in \mathcal{M}$ can be mapped to each other via the action of a group element $g \in G$ via $m_2 = g \cdot m_1$. E.g., the plane $\mathcal{M} = \mathbb{R}^2$ is a homogeneous space of the special Euclidean motion group $G = SE(2)$ as any two points can be mapped to each other through a rotation and a translation. Such groups $G$, which have $\mathcal{M}$ as a homogeneous space, can always be split in two parts via the semi-direct product $G = \mathcal{M} \rtimes H$, with $H$ a sub-group of $G$ that leaves some reference point $m_0 \in \mathcal{M}$ invariant, i.e., $\forall_{h \in H} : m_0 = h \cdot m_0$. E.g., rotations leave the zero vector in $\mathcal{M} = \mathbb{R}^2$ invariant, and thus $H = SO(2)$ in the $SE(2)$ case. Conversely, any homogenous space can be modeled with a group quotient $\mathcal{M} = G/H$.

We define an anisotropic manifold graph to be a discretization of a Lie group $G$ of which $\mathcal{M}$ is a homogeneous space. It consists of a finite set of vertices corresponding to a random sampling of group elements, and a finite set of similarity-based edges that are constructed via a left-invariant Riemannian metric on $G$. In our work we consider two anisotropic manifold graphs: one associated with the base manifold $\mathcal{M} = \mathbb{R}^2$ which we extend with an additional orientation/rotation dimension $H = SO(2)$ to come to the Lie group $G = SE(2) = \mathbb{R}^2 \rtimes SO(2)$, and the other associated with the sphere $\mathcal{M} = S^2$ which we similarly "lift" to the Lie group $G = SO(3)$ by adding an additional rotation dimension. Considering the similarity between the two cases (the sphere locally looks like $\mathbb{R}^2$) we will refer to $\mathcal{M}$ as the "spatial" part, and $H$ as the "orientation" part of the group.

**Uniform sampling of the vertices.**   The first step to construct an anisotropic manifold graph is to sample elements on the group uniformly or as uniformly as possible if the manifold does not permit a uniform grid. We split the grid construction in two parts, a grid on $\mathcal{M}$ which is sampled with $|\mathcal{V}_s|$ points and a grid on $H$ that is sampled with $|\mathcal{V}_o|$ points, leading to a total of $|\mathcal{V}| = |\mathcal{V}_s||\mathcal{V}_o|$ vertices.

**Left-invariant anisotropic Riemannian distance.**   Once vertices have been uniformly sampled on the group manifold, a similarity measure between vertices is computed. This measure is based on a Riemannian distance between points in $G$. The only thing one needs in our algorithm is the implementation of the logarithmic map on the Lie group (see e.g. [Bekkers, 2019]), and a diagonal Riemannian metric tensor (see e.g. [Sanguinetti et al., 2015] and [Mashtakov et al., 2017] for the $SE(2)$ and $SO(3)$ case respectively). In the following we provide the essential idea and intuition behind the construction of the similarity measure and provide a more extensive treatment in App. B.

In Riemannian geometry on Lie groups it is common to express tangent vectors of curves in a basis of left-invariant vector fields as it allows to measure their lengths with a single Riemannian metric tensor that is shared over the entire group. This works as follows. Consider curve $\gamma : [0, 1] \rightarrow G$ with its tangent vectors $\dot{\gamma}(t) = \sum_{i=1}^{d} u^i(t) \mathcal{A}_i|_{\gamma(t)}$ expressed in a basis/moving frame of reference

---

[2]Because spectral graph NNs are able to capture the geometry of the space, which in this work we equip with anisotropic metrics, any spectral method could be made equivariant using our method.

$\{\mathcal{A}_i|_{\gamma(t)}\}_{i=1}^d$, in which $\mathcal{A}_i$ are left-invariant vector fields. The length of these tangent vectors is then measured by a Riemannian metric tensor that we denote with $\|\dot{\gamma}(t)\|_{\mathbf{R}}^2 := \mathbf{u}(t)^T \mathbf{R} \mathbf{u}(t)$, with $\mathbf{R}$ a symmetric positive definite matrix defined relative to the basis $\{\mathcal{A}_i|_{\gamma(t)}\}_{i=1}^d$, and with $\mathbf{u}(t) = (u_0(t), u_1(t), \dots)^T$. The $\mathcal{A}_i$ are left-invariant vector fields and the notation $\mathcal{A}_i|_g$ means the vector in the vector field $\mathcal{A}_i$ at location $g$. The vector fields are constructed by choosing a vector $A_i$ in the tangent space at origin (the Lie algebra) which then defines a complete vector field on $G$ via the push-forward of left-multiplication. In less technical terms this means that if we pick a direction vector at the origin, and we move it to another point in, e.g. $G = SE(2)$, via a roto-translation, this vector will move and rotate along. By defining everything in terms of these left-invariant vector fields, every tangent space $T_g(G)$ at each $g \in G$ can be identified with the tangent space at the origin, and a single Riemannian metric tensor $\mathbf{R}$ can be shared over the entire space. Moreover, the induced Riemannian distance $d(g, h)$ between any two points $g, h \in G$ is then by construction left-invariant, i.e., $\forall_{g,h,i \in G} : d(g \cdot h, g \cdot i) = d(g, i)$.

Expressing tangent vectors in such left-invariant vector fields allows us to reason in terms of the generators of the group. Consider the $G = SE(2)$ case. As a basis we pick the 3 generators of the group: a forward motion represented by a vector $A_1$ pointing in the forward direction within the plane, a side-ways motion represented by a perpendicular planar vector $A_2$, and a rotation/change of orientation represented by a vector $A_3$ that points vertically in along the $H$-dimension. We then work with diagonal Riemannian metric tensors $\mathbf{R} = \text{diag}(1, \epsilon^{-2}, \xi^2)$, which penalize each type of motion (represented by the vector components) differently. When $\epsilon \to 0$ one arrives at the *sub-Riemannian geometry* which forms the basis for the mathematical modeling of visual perception. It quantifies a notion of alignment through the sub-Riemannian distance; the length of a distance-minimizing geodesic that connects two local orientations that lie in the extend of each other will be much smaller that that of a geodesic connecting two local orientations parallel to each other. An analogy can be found with the example of a car in a parking lot where it can move forward/backward ($A_1$) and change orientation ($A_3$) [Reeds and Shepp, 1990]. It will be easier to move it to the more aligned spot directly ahead then it will to the spot next to the car, as sideways motion ($A_2$) is impossible.

Parameters $\epsilon$ and $\xi$ will respectively be referred to as spatial and orientation anisotropy parameters. With $\epsilon = 1$ the metric is isotropic and there will be no distinction between different orientations. When $\epsilon < 1$, $\xi$ determines the flexibilty/curvature of the geodesics as it balances spatial motion against angular motion. In a sense it defines how easily one connects local orientations that are not optimally aligned. In Figure 1 this behavior is visualized by running a diffusion process on the anisotropic manifold graph. In the anisotropic case ($\epsilon < 1$) diffusion is faster along the forward direction within a $\theta$-plane. From a graph NN perspective this suggests that information is propagated more quickly between vertices that are aligned, nevertheless, Chow's theorem (see e.g. [Montgomery, 2006]) guarantees that any point pair in the (sub-)Riemannian manfiold can interact with one another.

The exact computation of the (sub-)Riemannian distances is challenging and can generally not be done in closed form, but can be done numerically via method such as [Bekkers et al., 2015, Sanguinetti et al., 2015, Mashtakov et al., 2017]. In order to keep our graph construction algorithm efficient though, we will approximate the Riemannian distances via an efficient analytic formula based on those in [Bekkers et al., 2018] that only involves the Lie group's logarithmic map $\log : G \to T_e(G)$ and the Riemannian metric tensor $\mathbf{R}$. We then approximate the distance between points $g, h \in G$ by

$$d(g, h) = d(e, g^{-1} \cdot h) \simeq || \log(g^{-1} \cdot h) ||_{\mathbf{R}}. \tag{3}$$

**Similarity measure.** Encoding a similarity measure in the edges of a graph requires defining a weighting scheme. It is common to use a Gaussian kernel and set the weights via

$$w(v_i, v_j) = \begin{cases} \exp\left(-\frac{d^2(v_i, v_j)}{4t}\right) & \text{if } e(v_i, v_j) \in \mathcal{E} \\ 0 & \text{otherwise} \end{cases}. \tag{4}$$

The choice for kernel bandwidth $t$ is essentially arbitrary, but good heuristics exist. Perraudin et al. [2019] set it to half the average squared distance between connected vertices. Defferrard et al. [2020], however, showed that this heuristic has the tendency to overestimate it and preferred to choose it as the minimizer of the mean equivariance error. Following this overestimation observation, we fix the kernel bandwidth as 20% of the average squared Riemannian distance between connected vertices. As such, the weights diversely cover values in the whole range $[0, 1]$. The most similar vertices are connected with close-to-one weighted edges whereas the lowest connections are close to zero.

**Quality of the approximation.** In theory, we would like our approximation to be as precise as possible. In practice, a high-resolution approximation leads to computational issues in time and memory. Hence, tuning of the graph parameters becomes a trade-off between theoretical consistency and practical feasibility. First of all, the graph resolution (or the number of vertices we sample) is directly related to the quality of the approximation. While the spatial resolution $|\mathcal{V}|_s$ is usually determined by the data (up to up- and down-samplings), the orientation resolution $|\mathcal{V}|_o$ is a design choice. An important remark is to notice that a large orientation resolution does not necessarily help if two different orientations are not distinguishable because of a poor spatial resolution [Weiler et al., 2018, Bekkers, 2019]. Secondly, the connectivity of the graph is also a crucial parameter. A fully connected graph is theoretically the best approximation. Nevertheless, for computational reasons, we use $K$-NN graphs[3] to sparsify the graph Laplacians.

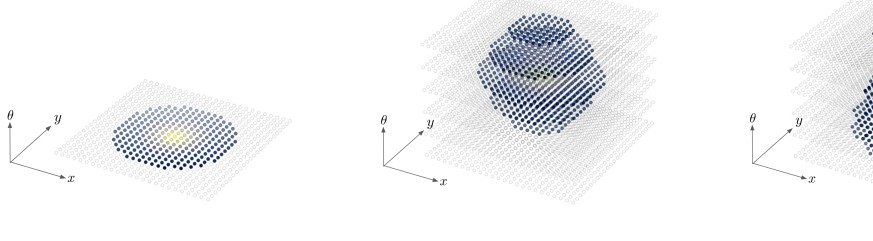

(a) Base space $\mathcal{M}$ with an isotropic Riemannian metric

(b) Lie group extension $G$ with an isotropic Riemannian metric

(c) Lie group extension $G$ with an anisotropic Riemannian metric

Figure 1: Isotropic diffusion applied to an impulse signal on Riemannian manifolds on $\mathcal{M} = \mathbb{R}^2$ and $G = SE(2)$.

**Theoretical group equivariance of the graph Laplacian.** Due to the success of machine learning algorithms based on graph Laplacian, the theoretical convergence of the graph Laplacian to its continuous analogue has been largely studied [Hein et al., 2005, Singer, 2006]. Belkin and Niyogi [2006] noticed that in many graph-based algorithms, a central role is played by the graph Laplacian's eigenvectors. Thus, they focused on proving convergence in eigenmaps as it is sufficient in this case. They proved that if the graph's vertices are sampled uniformly from an unknown submanifold $\mathcal{M} \in \mathbb{R}^d$, then the eigenvectors of a suitably constructed graph Laplacian converges to the eigenfunctions of the Laplace-Beltrami operator on $\mathcal{M}$. Consequently, as the latter operator is left-invariant, as we show in theorem A.1, the graph Laplacian is asymptotically [4] group equivariant.

**Empirical group equivariance of the graph Laplacian.** We empirically confirm the group equivariance property of the graph Laplacian applied to our anisotropic manifold graphs. By checking $P^\top \tilde{\Delta} P = \tilde{\Delta}$ where $P$ is a permutation matrix, we can verify that the graph Laplacian is invariant under a given permutation of vertices corresponding to a group transformation (e.g. a rotation of the graph). Moreover, we can also compare the eigenmaps of a graph Laplacian and its continuous counterpart if it is well-known. For a further discussion about this, see App. C.

### 3.2 ChebLieNet

**Chebyshev convolutional layer.** As introduced in Defferrard et al. [2016], a Chebyshev convolutional layer is a spectral layer based on a continuous kernel parametrization with graph Laplacians. This parameterization makes such layers highly suitable for our method, as they intrinsically capture the Riemannian geometry of the graphs on $G$. Moreover, the Chebyshev convolutions on the anisotropic manifold graphs are equivariant by construction because the graph Laplacians are equivariant operators (see Figure 2).

---

[3]Note that in our implementation, a $K$-NN graphs does not mean that each vertex has $K$ neighbors but at most $K$ neighbors. Indeed, if the graph domain has boundaries, using exactly $K$ neighbors for each vertex could lead to asymmetries that may introduce biases and harm the permutation invariances in the graph.

[4]The asymptotic case corresponds to $|\mathcal{V}| \to \infty$ and a Gaussian weight kernel with kernel bandwidth $t \to 0$.

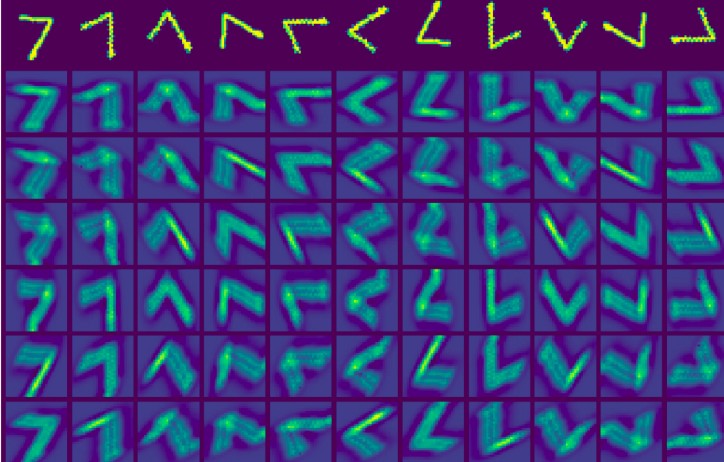

Figure 2: Rotation equivariance of a randomly initialized $SE(2)$ Chebyshev convolutional layer. From left to right shows different rotations of an input (top row) and the activations for different slices of $\theta \in [0, \pi]$ in the graph (bottom 6 rows). A rotation of an input image followed by Chebyshev convolution is equivalent to first convolution followed by a planar rotation in each $\theta$ slice and a roll in the $\theta$-axis.

**Spatial pooling and unpooling layers.** Graph pooling is a central component in a myriad of graph neural network architectures. Producing coarsened graphs from a finer graph have two main advantages: first, it reduces the computational cost, and second, it could improve performance by reducing the overfitting effect and adding a multiscale perspective. As an inheritance from traditional CNNs, most approaches formulate graph pooling as a cluster assignment problem, extending local patches' idea in regular grids to graphs [Dhillon et al., 2007, Ying et al., 2018, Khasahmadi et al., 2020, Mesquita et al., 2020]. We propose similar operations on the base space (spatial domain) and involving two steps (see Figure 3). First, each sample is assigned to a cluster that will correspond to the output sample; this is the down- (resp. up-) sampling phase. With a well designed method, this change of data-resolution can be made equivariant to any group transformation.[5] Then, each cluster is reduced (resp. expanded) according to a given scheme (e.g. maximum, average or random); this is the reduction (resp. expansion) phase. When the reduction and expansion steps are permutation-invariant operations, such layers are automatically invariant under any transformation in the group.

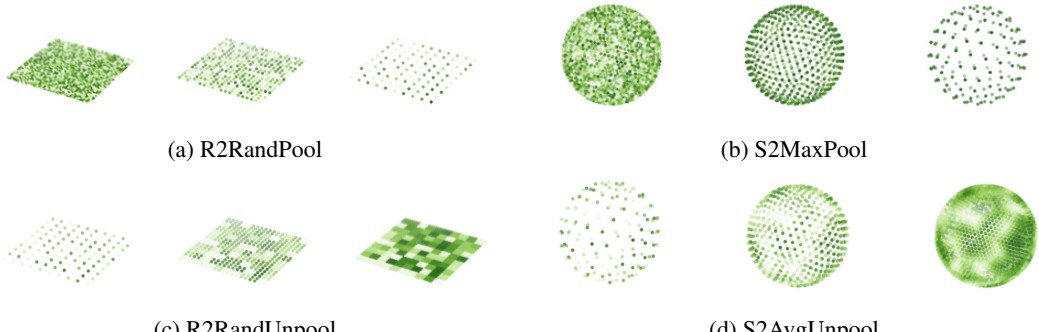

(a) R2RandPool                                    (b) S2MaxPool

(c) R2RandUnpool                                 (d) S2AvgUnpool

Figure 3: Spatial pooling and unpooling layers on the 2D grid and the sphere.

**Global pooling (projection) layer and point-wise operations.** When the neural network does not need to be equivariant but invariant (e.g. classification task), it is common to rely on a global pooling layer (or simply projection layer). This layer reduces the d-dimensional signal on the graph's vertices to a d-dimensional vector of features derived from information on the whole graph. As a permutation-invariant operation, such a layer does not break the equivariance property of the neural network. Finally, point-wise operations do not affect the equivariance of a neural network.

---

[5]Altough down- and up-samplings are naturally defined on the Euclidean grid, this task is more complicated on the sphere. However, using an icosahedron decomposition of the sphere, we make it more natural as down- and up-sampling consists of decreasing or increasing the subdivision level.

# 4 Experiments

In this section, we show the benefits of working on the anisotropic manifold graphs compared to the base manifold graphs. We believe that further improvements could be achieved through tuning and hyper-parameter optimization of the models [Yu and Zhu, 2020], using high-capacity networks, or via a more advanced training process, but this is not the goal of our work. We here intent to illustrate the adaptability of our approach to different tasks such as classification and segmentation in 2D images or spherical data. In the first couple of experiments, we motive the use of anisotropic spaces. By varying the anisotropies, we show the existence of sweet spots, both for the spatial anisotropy parameter $\epsilon$ and the orientation anisotropy parameter $\xi$. In the second couple of experiments, we show that even if we add a new orientation dimension, our method remains scalable using a proper implementation.

Our implementation is fully PyTorch [Paszke et al., 2019] and available at `https://anonymous.url`. We perform all the experiments on a single GeForce GTX 1080 Ti gpu and track them with the Weights & Biases library [Biewald, 2020]. The details of the experiments are given in the App. D.

## 4.1 Why using tunable anisotropic kernels?

As introduced in Section 3.1, the anisotropies are tunable via the parameters $\epsilon$ and $\xi$ of the Riemannian metric, respectively responsible for the spatial and orientation anisotropies. As the $\xi$ parameter should depend on the spatial and orientation resolutions, we use the following parameterisation: $\xi^2 = \alpha \frac{|\mathcal{V}_o|}{|\mathcal{V}_s|}$. Setting $\alpha = 1$ yields a 40/60 ratio of neighbors within versus outside the orientation plane. We ran different experiments with a Wide Residual architecture [Zagoruyko and Komodakis, 2016] on CIFAR10 [Krizhevsky et al., 2009], varying the spatial and orientation anisotropic parameters.

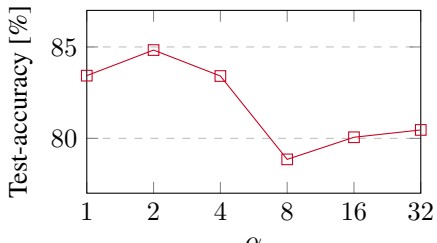

(a) Test-accuracy against orientation anisotropies

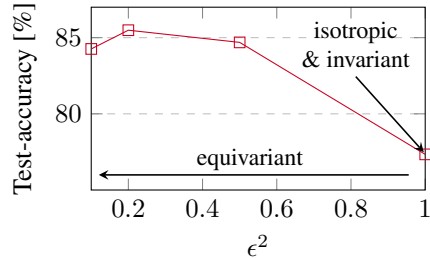

(b) Test-accuracy against spatial anisotropies

Figure 4: Empirical proof of existence of sweet spots for data-dependent anisotropic parameters.

**Orientation anisotropy.**  The orientation anisotropy $\xi$ controls how strongly orientation layers are connected. At the limit $\xi \to \infty$, orientation layers are decoupled. It is like test-time augmentation with rotations: running a CNN working with one anisotropic Laplacian (e.g., only vertically aligned filters) and testing the network for different input rotations before averaging the output. The other extreme $\xi \to 0$ keeps all layers equally close to each other, and features are essentially identified with just a spatial coordinate. This would then correspond to a WideResNet with isotropic Chebyshev convolutions. For reasonable values of $\xi$, interactions between orientation layers take place. Figure 4a is evidence of the existence of a sweet spot for this parameter in the range of reasonable values. At the moment, we expect with no certainty that this parameter could be set *a priori* of the data, only considering the data resolution. As a rule of thumb, we set $\xi$ such that each vertex has approximately 40% of its neighbors in the same orientation layer and 60% on others.

**Spatial anisotropy.**  The spatial anisotropy $\epsilon$ regulates the anisotropy of the space on the spatial domain. For $\epsilon = 1$, the Riemannian metric is spatially isotropic; all directions are treated equally and the resulting model would effectively be a WideResNet with isotropic Chebyshev convolutions. At the limit $\epsilon \to 0$, the main direction has a minimal cost, and the resulting space is highly spatially anisotropic. In figure 4a we observe that using anisotropic spaces instead of isotropic ones is relevant, as we almost get an 8% test-accuracy improvement. Unlike the orientation anisotropic parameter, in our opinion, this parameter is task/data-dependent; different datasets could benefit in different degrees from the utilization of directional information through different spatial anisotropy settings.

## 4.2 How scalable is the method?

Scalability is often an important limitation of graph- and group-based neural networks. By adding an orientation dimension, we do not run from this rule as we necessarily increase the number of vertices of the anisotropic manifold graphs. To permit experiments on larger images, it becomes crucial to pre-compute anisotropic manifold graphs and their Laplacians. Dedicated libraries like PyKeops [Charlier et al., 2020] enable this without memory issues. Nevertheless, the graph operations (convolutions, pooling or unpooling) still scale with the size of the graph. Fortunately, PyTorch provides sparse operations that increase efficiency in terms of time and memory compared to dense operations in cases of sufficiently sparse graph Laplacians (typically a sparsity $\mathcal{S}(\tilde{\boldsymbol{\Delta}}) \geq 98.5\%$).

We evaluate our models on an image classification task on STL10 [Coates et al., 2011] and an image segmentation task on ClimateNet [Kashinath et al., 2021]. We show the adaptability of our method by using a Wide Residual architecture [Zagoruyko and Komodakis, 2016] on STL10 and a U-Net-like network [Ronneberger et al., 2015] on ClimateNet. We also demonstrate the potential of our approach and the benefits of using anisotropic spaces. Indeed, while on ClimateNet the use of anisotropies is neither beneficial nor detrimental, the difference in performance on STL10 is significant.

Table 1: Mean of test performance and training duration on ClimateNet and STL10. Errorbars are 1 standard deviation computed over 5 trials.

| | | ClimateNet | | STL10 | |
|---|---|---|---|---|---|
| $\epsilon$ | | Test F1 | Duration | Test accuracy | Duration |
| 1 | (invariant) | $\mathbf{85.62 \pm 0.09\%}$ | $\sim 2\,\mathrm{d}$ | $68.98 \pm 0.56\%$ | $\sim 9\,\mathrm{h}$ |
| 0.1 | (equivariant) | $85.25 \pm 0.19\%$ | $\sim 7\,\mathrm{d}$ | $\mathbf{74.02 \pm 1.10\%}$ | $\sim 16\,\mathrm{h}$ |

## 5 Conclusion

**Scope.** With our method, geometric graph NNs are made equivariant to Lie groups. Via the groups $SE(2)$ and $SE(3)$, we can construct roto-translation equivariant networks for $2D$ image data and $3D$ volumetric data. Based on the group $SO(3)$, our method can deal with meteorological or cosmological data while preserving rotation equivariance. We believe that our flexible approach is ideal for further explorations on the relevance of group equivariance in tasks not considered in this work.

**Limitations.** The main weakness of our method is its relatively high memory requirement. Although all experiments ran on a single gpu, by adding an orientation axis, we significantly enlarge the feature maps. As a result, anisotropic graph manifolds are memory-heavier than isotropic ones and prone to a slowdown during the forward- and backward-pass. Nevertheless, with the emergence of geometric deep learning, we expect improvement in the hardware and implementation of graph-oriented operations. Another challenge is the increased number of hyper-parameters for which we only have derived rules of thumb. The graph connectivity and resolutions require a tradeoff between efficiency and quality of the manifold approximation. The anisotropic parameters require an analysis of the dataset and some intuition about the amount of anisotropy to set. With systematic hyper-parameter optimization, we can find an optimal combination, but requires more computational resources.

**Potential and future research.** Thanks to its easy-to-tune anisotropic properties, our model can be used to better understand anisotropic properties in data. In particular, one could explore the effect of using anisotropic spaces instead of isotropic ones on many tasks and conclude when such anisotropic information is relevant. In this vein, it could also be interesting to derive anisotropic pooling and unpooling layers based on anisotropic spaces instead of isotropic ones as it is usually done. More generally, our method is simple enough to be extended to shapes/surfaces with a Riemannian manifold structure [Cohen et al., 2019]. In this work, we focused on 2D images and spherical data on, but the method is readily extendable to higher dimensional Lie groups such as the $SE(3)$ group to obtain 3D roto-translation equivariant ChebLieNets. Moreover, our method for constructing anisotropic geometries could directly improve other successful Euclidean distance-based graph NNs such as [Satorras et al., 2021] by making them fully equivariant. Last but not least, despite graph-based algorithms being computationally sub-optimal compared to CNNs, their flexibility is a real asset. We see high potential in the exploration of graph sparsification to reduce computational complexity.

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
