# OpenReview forum: "ChebLieNet: Invariant spectral graph NNs turned equivariant by Riemannian geometry on Lie groups"
_NeurIPS.cc/2021/Conference — NeurIPS 2021 Submitted_

### Official Review · Reviewer_hWjk · 2021-07-12

**Rating:** 5
**Confidence:** 3

**Summary:**

Authors introduce graphs equivariant convolutional neural networks that are group-equivariant. They build on Defferrard et al. 2017, which introduced scalable convolutional neural networks on graphs, leveraging Chebyshev polynomials to bypass the need to compute the Fourier basis. In particular, they lift the (homogeneous) base space on which the data is defined to a Lie group (which acts on the base space) where they parametrize an anisotropic Riemannian metric. The edges weight are then computed using the geodesic distance induced that the metric, reflecting anisotropy.


**Limitations And Societal Impact:**

Limitations are discussed.

**Main Review:**


## Strengths
This work is relevant to the community as it contributes to building models that encode inductive biases in the form of group symmetry. The authors seem well versed in Riemannian geometry, group theory and spectral theory.
Empirically, they show that incorporating both 'spatial' and 'orientation' anisotropy for CIFAR10 helps on generalization performance.

## Weaknesses
The main weakness of this work perhaps is clarity. I expand on this below.
Additionally, empirical evidence backing up the motivation of this work is somewhat limited. For instance, on Figure 4, the best test accuracy is still quite low, WideResNet reported an an accuracy of 96.11%. (I may be missing something). Similarly for STL10. I would argue to include a simple synthetic dataset with anisotropy baked in, and showing some of the learned filters so as to help the reader understand the method better.

## Clarity
Generally the paper needs some improvement with regards to clarity. The method is grounded in a mathematical framework, which some readers may only be partially familiar with. As a consequence, particular care is required when introducing core ideas. For instance, it is somewhat stated that an anisotropic metric leads to the equivariance of the graph convolution layers whereas an isotropic metric leads to invariance. As this seems key I would suggest to expand and clarify this.
As section 3 is where the method is formally introduced, it needs to convey efficiently the ideas that the method builds on. I believe that the paragraph on local frames and co could be moved to the appendix as it is not adding much. The paragraphs that follow are key as they introduce the parametrization of the Riemannian metric with the spatial and orientation anisotropies. I believe that a bit more work on these is necessary to give better intuition to the reader. Perhaps an illustration would help? Figure 1 is helping but still not filling the gap.

## Correctness
This work claims that the proposed ChebLieNet is group-equivariant as graph Laplacians are equivariant operators. This is true as the Laplacian is intrisic, hence invariant under isometries. As this is a core claim of the paper, it would nonetheless  be useful for the reader to have a bit more explanations and intuition.
The convolution is in practice only approximately equivariant due to the Laplacian being asymptotically equivariant. How come this requires the kernel bandwidth to go to infinity? Wouldn't all the weights go to zero then?

## Reproducibility
The code source has been provided in the supplementary material and after a quick inspection, it seems to allow to reproduce their results.

## Additional feedback
- As the base space is lifted, how is the $H$ dimension (grid) initialized for the data?
- 197: "which forms the basis for the mathematical modeling of visual perception." This sentence would need more context, or could be simply removed.
- 218: Can we really not analytically compute the distance the geodesic distance in $SE(3)$? Is it because of the particular choice of Riemannian metric R? Does the Lie group logarithm map depends on $\epsilon$ and $\xi$? The approximation from Equation 3 linearize the group around $h$ (or symmetrically $g$) which sounds reasonable as the filter is localized. This may be highlighted.
- Figure 1: Perhaps adding a 4th subfigure `Base space M with an anisotropic Riemannian metric' would help?


**Time Spent Reviewing:**

4

---

> ### Author Response · Authors · 2021-08-11
> **Response to Review of Paper9906 by Reviewer hWjk**
>
> Thank you for your time and thoughtful comments.
>
> Regarding the weakness:
> We thank you for the nice recommendation of working with a synthetic dataset with anisotropies baked in and visualizing the filters. This is a great idea, however, when it comes to constructing anisotropic dataset it is a bit complicated. Namely, anisotropies would refer to local structures/patterns and in a way this is already the case in the MNIST dataset and the other datasets. For example MNIST digits are composed of local line segments, which are anisotropic structures. So we think we can already make the method insightful by visualizing filters learned on these datasets. We will do so with a new figure that shows for the method acting only on the base space M the filters are indeed isotropic and therefore can only detect isotropic patterns, whereas in the lifted/anisotropic case the filters are anisotropic and can detect elongated patterns.
>
> Then when it comes to SOTA results: we are however not aiming for nor claiming SOTA results (neither on Figure 4 nor for STL10). We refer the reviewer to our top-level "General Response" comment, where that question is further addressed.
>
> Regarding clarity:
> Many thanks for your suggestions on how to improve clarity. We will take them into account to revise the manuscript.
> In particular, we will clarify how anisotropic and isotropic Riemannian metrics lead to equivariant and invariant convolutions, and add explanations about graph Laplacians being equivariant operators.
>
> We will furthermore make an improvement of Figure 1 to better connect it to the text.
>
> Regarding the correctness remark:
> The convolution is indeed only approximately equivariant as the Laplacian is asymptotically equivariant. The graph Laplacian converges asymptotically when the kernel bandwidth goes to 0 *and* the number of vertices goes to infinity. Intuitively, the kernel has to concentrate to compensate for the increasing number of vertices falling in a given ball.
>
> Regarding your additional feedback:
> * We are not sure if we understood the question correctly, but perhaps the following clarification helps. As the base space is lifted, the H dimension can be initialized with an arbitrary discretization (or number of orientation layers). As the sub-groups are 1 dimensional the discretization is a uniform grid over 1D rotations (the sub-group H).
> * 197: The context is introduced in lines 29-51, with the neuroscientific motivation of using SR geometry to develop new deep learning algorithms. See also our response to Reviewer MH5E (point 4). We will insert appropriate references here.
> * 218: We cannot compute the distance in closed form. The problem is that the Riemannian metric tensor changes at each location in G (though relative to a moving frame of reference it appears to be constant). Were the metric to be isotropic or constant everywhere the log-norm would provide the exact distance, however now it only locally is an accurate approximation. Note that the usual geodesics (the exponential curves) would be straight curves, or in fact circular spirals in SE(2), but in the sub-Riemannian setting they have more complex shapes as visualized in [this figure](https://www.dropbox.com/s/h7pdo9cnk6l6haz/Screen%20Shot%202021-08-11%20at%2009.29.00.png?dl=0). Then, the linearization of the group around h is indeed reasonable because Chebyschev filters are localized. We will highlight that.
> * Figure 1: This is indeed a good way to show that one needs to lift to a higher dimensional manifold. Namely, when sticking to the base space M one has to pick one out of many possible anisotropy directions, which create a bias towards that selected orientation. By lifting one can consider all directions indexed by the additional axis. We can then connect this updated figure with the visualization of the filters in the new figure that we will include in the revision.

---

> > ### Comment · Reviewer_hWjk · 2021-08-18
> > **Reply to rebuttal**
> >
> > I thank the authors for the clarifying comments. In particular regarding the asymptotic convergence of the Laplacian and the non-availability of closed-form formula for the geodesic distance in $SE(3)$.
> >
> > Nonetheless, I believe that the lack of clarity of the current submission is too damaging for the paper to be accepted in its current form.
> > For this reason I stand by my original rating and encourage the authors to improve their submission following the reviewers recommendations.

---

### Official Review · Reviewer_MH5E · 2021-07-14

**Rating:** 5
**Confidence:** 5

**Summary:**

In the paper, a new graph neural network called ChebLieNet was proposed. The proposed network was designed to achieve equivariance properties with respect to actions of Lie groups. The proposed ChebLieNet was analyzed on vanilla datasets.

In summary, the proposed ChebLieNet was well motivated and structured. However, there are various major and minor problems with the paper including missing notation, definitions, incomplete and insufficient mathematical results and experimental analyses, especially in comparison with the state-of-the-art.

After Rebuttal: Thank you for your response to my comments. I checked the comments of other reviewers and the response to their comments as well. The authors answered most of the questions in the rebuttal. However, the paper should be improved for a clear acceptance. Therefore, I slightly increase my grade, and recommend authors address the theoretical details and improve the experimental analyses, esp. in comparison with state-of-the-art related methods..

**Limitations And Societal Impact:**

Partially. Please check the above comments.

**Main Review:**

Comments and suggestions on major and minor issues with the paper:

1.	There are various unclear statements in the paper. For instance, even the first sentence in the abstract “We introduce ChebLieNet, a group-equivariant method on (anisotropic) manifolds” is not clear. More precisely, is ChebLieNet a method, algorithm or a new neural network? What do you mean by “group-equivariant method”? The title should be also updated with a more clear description.

2.	There are also various undefined terms and notation (I also checked supp. mat.). For instance, definitions of invariance/equivariant isotropic/anisotropic metric/manifold/layer/network/method are missing and these terms are used interchangeably without considering terminological and mathematical precision. Therefore, a reader needs to make educated guess on various terms, and the paper is not easily readable and most of the parts should be rewritten. Please also provide definition of the notation as it is used (e.g. for eqn. (1)).

3.	Most of the claims proposed in the paper should be revised since they are not verified mathematically or experimentally. Note that, the mathematical statements given in the paper construct the formalism of the proposed method/network but not introduces mathematical/theoretical results supporting the strong claims. For instance, the following claims should be explored and analyzed more precisely:

a.	With our method, geometric graph NNs are made equivariant to Lie groups: Needs mathematical and experimental verification. This property is not proved and detailed experimental analyses were not given.

b.	We demonstrate the equivariance property of ChebLieNet, both in theory and in practice. This property guarantees that the neural network’s predictions are robust against given transformations, which is not necessarily the case with methods based on data augmentation: Needs mathematical and experimental verification: Needs mathematical and experimental verification. This property is not proved and detailed experimental analyses were not given. You need to provide results and analyses at least for different transformations and data augmentation methods in comparison with state-of-the-art methods.

c.	We show that the use of directional information via anisotropic Riemannian spaces could benefit many tasks: Needs mathematical and experimental verification. This property is not proved and detailed experimental analyses were not given. You need to provide results and analyses at least for 4-5 different tasks.

d.	We show the flexibility of the method by considering two different problems; we validate on classification problems with 2D image data and a segmentation problem on spherical data via the construction of a sub-Riemannian geometry on SE(2) and SO(3) respectively. Needs more detailed experimental verification. If you would like to address segmentation problem, please also consider the problem on image data.

4.	Please explain more clearly how “the sub-Riemannian geometry forms the basis for the mathematical modeling of visual perception.”

5.	The experimental analyses should be improved also by comparison of the proposed method with state-of-the-art invariant and equivariant methods and network models on different datasets including benchmark graph datasets.


**Time Spent Reviewing:**

3

---

> ### Author Response · Authors · 2021-08-11
> **Response to Review of Paper9906 by Reviewer MH5E**
>
> Thank you for your time and thoughtful comments. In the following we respond to your expressed concerns. Overall, based on your valuable feedback we believe that the paper can significantly improve from textual improvements, focusing on overall clarity of presentation as well as more intuitive and accessible introduction to the technical aspects of the paper.
>
> 1. On the categorisation of ChebLieNet as a method. The main idea of the paper is to present a method that could be used with any spectral graph NNs on an anisotropic extension of the original isotropic base space. The term method refers then to a general idea to turn equivariant an initially invariant neural network. That is, spectral methods are not able to exploit directional or anisotropic cues in the data as the Laplacian is a (rotationally) invariant operator. By modifying the underlying geometry of the graph we can however exploit directional information whilst using the same spectral method. We must admit that the main idea could have been more explicitly and intuitively presented and agree that the title and abstract are rather technical and require additional explanation. We will increase the accessibility for our revision with additional explanations of technical terms to make it self-contained. We do prefer, however, to keep the title as it is, even considering technical terms such as "equivariance". As we see an increase of papers on equivariant deep learning and an increasing interest of the field in the topic, we believe that is appropriate.
>
> 2. Regarding undefined terms. Thank you for bringing this to our attention. We see great value in addressing this readability issue and see how it improves the paper. We will make sure all technical terms are explained and double-check for potential ambiguities.
>
> 3. Regarding mathematical and experimental validation of claims. We agree that most of the effort of the paper goes to formalising the method itself, followed by an empirical validation rather than a theoretical validation. Our objective is to present a new method to be used in practice, and in order to describe it accurately, we need the math. While our initial objective was not an in-depth theoretical analysis but rather an investigation of its potential in applications, we see that one does not exclude the other. Thank you for pointing out properties of the method that can theoretically be verified. We will include this in the revision where possible, see below.
>
> a. and b. Equivariance of the method is guaranteed by construction. The graph Laplacian is theoretically known to converge to its continuous counterpart, the Laplace-Beltrami operator. As remarked by Reviewer hWjk, the Laplace-Beltrami is intrinsic, hence invariant under isometries, and thus group-equivariant. As shown in Figure 2 for SE(2), a convolutional layer is group-equivariant. We also empirically demonstrate this property on a classification task where the performances on MNIST and a randomly rotated version of it are very close to each other (see Figure 7). Empirical and theoretical convergence are discussed in 3.1 and in the appendix.
>
> c. Anisotropic convolutions could benefit many tasks. We believe the criterion for showing the results for at least 4-5 tasks is unreasonable, and stick to a motivation for why we think the statement holds. We will underpin this statement with references to the equivariance literature as all papers on group convolutions (from the seminal paper by Cohen and Welling onwards) show that exploiting directional cues via group-equivariant operators is beneficial in a wide range of tasks. This can be argued in terms of degrees of freedom: plain NNs are unconstrained > convolutional NNs are constrained to be translation equivariant > G-CNNs are constrained to be equivariant to larger groups > and then we have convolutional NNs with the constraint that filters be isotropic. The spectral convolutions considered in this paper are of the latter type and it seems that they may be overly constrained. However, depending on the application, it may be the right inductive bias.
>
> d. We agree that more experiments would improve the paper. However, we do not believe the paper improves to the extent it out weights the added work. Namely, in many recent works on group equivariant deep learning, it is shown that group-equivariant layers improve over the standard pure translation equivariant layers in both segmentation and classification tasks. The main focus of the experiments is to show how we can benefit from using directional information with initially invariant spectral graph NNs.
>
> 4. Regarding briefness of the connection to the visual cortex. We do provide a high-level discussion on this connection in the introduction. However, we can imagine that this does not provide a full explanation but rather motivation and directions for further reading. In section 3, when this notion is revisited in terms of sub-Riemannian geometry, we will reiterate the references provided in the introduction that underpin the statement of SR-geometry modelling visual perception. We consider going deeper than what is explained in the introduction to be beyond the scope of this paper.
>
> 5. Regarding lack of SOTA results. The purpose of this paper is not to engineer SOTA architectures but rather to present a new method for building architectures inspired by the visual system and the flexibility of graph NNs. We, therefore, decide to limit the scope on the newly introduced method and its properties.

---

> > ### Comment · Reviewer_MH5E · 2021-08-25
> > **Reply to rebuttal.**
> >
> > Thank you for your response to my comments. I checked the comments of other reviewers and the response to their comments as well. The authors answered most of the questions in the rebuttal. However, the paper should be improved for a clear acceptance. Therefore, I slightly increase my grade, and recommend authors address the theoretical details and improve the experimental analyses, esp. in comparison with state-of-the-art related methods.

---

### Official Review · Reviewer_JM2z · 2021-07-16

**Rating:** 6
**Confidence:** 1

**Summary:**

The authors introduce ChebLieNet as a group-equivariant method on manifolds that extends the original ChebNet. The extension is based on enforcing group equivariance via encoding anisotropic left-invariance Riemannian distance on the edges.

**Limitations And Societal Impact:**

The authors addressed the limitations and potential negative societal impact in Section 5 of the paper.

**Main Review:**

**Quality**: I checked most of the mathematical details from the paper. Except for a small typo on the left-invariance definition on line 190, all other details seem to be correct.

**Clarity**: The paper is in general very clear. The introduction and related work sections are written nicely, and the authors did a good job introducing the various components of their proposed network.

**Originality and Significance**: My research area is essentially disjoint from the area of the paper, and I do not have the expertise to comment on the originality or significance of the paper.

**Time Spent Reviewing:**

6

---

> ### Author Response · Authors · 2021-08-11
> **Response to Review of Paper9906 by Reviewer JM2z**
>
> Thank you for your time and thoughtful comments and thanks for spotting the typo, this will be fixed!

---

### Official Review · Reviewer_9mff · 2021-07-23

**Rating:** 4
**Confidence:** 3

**Summary:**

This paper proposes a learning method for graphs using equivariance for Lie groups on manifolds.
Specifically, it is an equivariant graph Laplacian-based neural network based on Lie groups equipped with an anisotropic Riemannian metric.
They have also provided some experiments for this approach.

**Main Review:**

There are a few things that I think are not good.
The first one is what kind of functions this model is a universal approximator for.
For example, in Maron's model, the universality of invariant continuous functions is known for the induced action of node permutation on adjacency matrices. Such a theoretical analysis is needed for this model.
I also have a question about discrete approximations of the Lie group: as the dimension of the Lie group increases, this kind of approximation becomes very computationally expensive, as seen in Lie conv. Is it always possible to approximate the Lie group to the level of practical accuracy with this method?
My last is about the experiment.
In Table 1, there are only results for the proposed method and no comparison method.
In this table, it is unclear whether the accuracy of the proposed method is higher than that of the previous methods or not.


**Time Spent Reviewing:**

10hours

---

> ### Author Response · Authors · 2021-08-11
> **Response to Review of Paper9906 by Reviewer 9mff**
>
> Thank you for your time and thoughtful comments. We identified three main concerns which we address as follows.
>
> The first is a question on the universality properties of the method. Firstly, we apologise for not being able to accurately convey the scope of the paper, and perhaps for misunderstanding your concern. Our method is not about proving new universality results, nor did we deem it necessary to prove statements about this for the proposed method at the time of submission as the main innovation of the method is in the construction of geometric graphs rather than layers that operate on them. However, we do see value in making statements about approximation power and are currently looking into how this can be effectively presented. The method builds upon a class of spectral methods for graph neural networks. The layers themselves, Chebyshev convolutions, are left untouched but is rather the geometry of the graph on which the method operates that is modified. Namely, the considered graphs are assumed to represent data on a homogeneous space that we extend to an anisotropic Riemannian geometry on a Lie group. The properties of spectral convolutions are otherwise untouched and we assume that universality results are inherited. We will put effort in formalising this based on the mentioned work of Maron and related works.
>
> On an additional note,  intuitively we believe that we increase universality by lifting from the (homogeneous) base space to the Lie group based on the intuition that anisotropic filters are more universal/powerful/general than isotropic ones, making the convolution equivariant rather than invariant to rotations. See also the degrees of freedom argument below. Then, equivariant functions are the most general functions that commute with the action of a symmetry group, see e.g. [Thm 1, Bekkers 2019] [Cohen et al. 2018] [Kondor, Trivedi, 2018].
>
> The second remark is on accuracy in the context of discretization. The method is akin to other group equivariant methods and similarly suffers from dimensionality issues. In general, denser, more compute-heavy discretizations reduce equivariance errors and improve performance. However, our method is not tied to a specific input grid due to the graph NN approach, which increases flexibility for dealing with computation costs, e.g., working with downsampled graphs (see e.g. Figure 3), random sparse grids or sparse adjacency matrices. Our method allows a precise control of the cost-performance tradeoff.
>
> The third remark is on a lack of comparison to other methods. Many different methods and architectures could have been chosen. However, a fair comparison between methods would then depend on how much engineering effort went into each approach to get the most out of the dataset. In turn, it complicates drawing reasonable conclusions from the experiments. Given that our approach could be used as a plugin replacement for classical graph NN building blocks, it made us most sense to choose a baseline architecture using common graph NN layers (Chebyshev convolutions) and then use the exact same architecture but replacing the underlying graph with a lifted anisotropic Riemannian manifold graph. By doing so, we could compare our proposed method fairly against a conventional baseline while using the same amount of parameters and the same architecture. It allows gaining insight into the performance of the method.

---

> ### Comment · Area_Chair_Hd5w · 2021-08-25
> **Please respond to author comments**
>
> Dear Reviewer 9mff,
>
> The authors have provided comments in response to your review. Has anything in their response changed your mind or caused you to increase your score?

---

### Author Response · Authors · 2021-08-11
**General Response**

We want to thank all the reviewers for their time, thoughtful comments, and valuable suggestions for improving the paper. We were happy to see that the reviewers considered that the "introduction and related work sections are written nicely" (JM2z), that our "proposed ChebLieNet was well motivated and structured" (MH5E) and that our "work is relevant to the community" (hWjk).

However, we also distilled the following main critiques from the reviewers which we could leverage to improve the paper. Here we respond to the main critiques on a general level and respond in detail to the individual reviews in their respective threads.

1. **Clarification.** From the reviews, we note that some parts of our work need to be clarified. We will make an effort in our revision to rectify the missing notation and definition and give a more precise and intuitive presentation of our method and its core components. We will furthermore improve the figures and insert a new one on the visualization of anisotropic filters. All reviewers gave excellent clues on where to make improvements in this direction.

2. **Comparison with the state-of-the-art.** We understand that more experiments with the state-of-the-art would make our analysis more rigorous. However, the objective of this paper is to present a method that allows standard graph NN layers (Chebychev layers specifically in this paper) to exploit directional information and anisotropies in the data, where on the original graphs they could not. I.e., our proposed method turns invariant methods into equivariant methods by construction of anisotropic geometric graphs. We found that empirically proving the necessity of using directional information was more important than engineering the neural network architecture to make them better than the current state-of-the-art. We consider our method to be a tool that can be used to push the state of the art, that, however, first needs to be validated for its properties. This is what we did in this paper and we hope that the reviewers appreciate this sentiment. We evaluated various neural networks with different degrees of anisotropies and showed that they could benefit from extending the initial isotropic base space to an anisotropic one.

---

### Decision · Program_Chairs · 2021-09-27

**Decision:**

Reject

**Comment:**

The authors present a novel type of layer for neural networks on manifolds, which can take into account the inherent anisotropy of the manifold. There was a consensus among reviewers that the paper lacked clarity and the presentation could be improved. The one reviewer who recommended acceptance only wrote a very cursory review and admitted that the paper was far outside their expertise, and so I have not taken their review into account. Therefore, I recommend the paper be rejected.